# Improved Phenolic Compositions and Sensory Attributes of Red Wines by *Saccharomyces cerevisiae* Mutant CM8 Overproducing Cell-Wall Mannoproteins

**Phoency F.-H. Lai** [1] , **Po-Chun Hsu** [1] , **Bo-Kang Liou** [2] , **Rupesh D. Divate** [1] ,
**Pei-Ming Wang** [1] **and Yun-Chin Chung** [1,*]

[1] Department of Food and Nutrition, Providence University, Taichung 433, Taiwan;
plai856@hotmail.com (P.F.-H.L.); ee41405@gamil.com (P.-C.H.); rupesh.divate@gmail.com (R.D.D.);
pmwang@pu.edu.tw (P.-M.W.)

[2] Department of Food Science & Technology, Central Taiwan University of Science and Technology,
Taichung 406, Taiwan; 106755@ctust.edu.tw

\* Correspondence: ycchun@pu.edu.tw; Tel.: +88-64-2632-8001 (ext. 15345)

**Abstract:** The objective of this study was to improve the quality attributes of red wines by *Saccharomyces cerevisiae* (BCRC 21685) mutant CM8 with overexpression of high-mannose mannoproteins, with respective to phenolic compositions, colorimetric parameters, and consumer sensory attributes. The CM8 was mutated by ethyl methane sulfonate and showed the ability of overproducing cell wall mannoproteins selected by killer-9 toxin-containing YPD plates. Kyoho grapes were used as raw materials. It is interesting to find that the cell wall mannoproteins isolated from CM8 mutant possessed a significantly higher mannose content in the polysaccharide fraction (81% *w/w*) than that did from parent strain (66% *w/w*). The red wines made of winter grapes and CM8 (CM8-WIN) showed significantly greater total tannins, flavonols, and anthocyanins levels, as well as higher color, higher flavor, and higher consumer preference than those by its SC counterpart (SC-WIN). The characteristics of the red wines studied were further elucidated by principal component analysis. Conclusively, using CM8 starter could effectively endow the red wine with high-quality attributes via the interactions of high-mannose mannoproteins with wine compounds.

**Keywords:** kyoho grape; mannoprotein; *Saccharomyces cerevisiae*; wine; consumer preference

## 1. Introduction

Yeasts are the critical microorganisms for winemaking, especially the strains of high ethanol productivities, such as *Saccharomyces cerevisiae*, *S. ellipsoideus*, and *S. pastorianus* [1]. During wine making, yeast cell wall mannoproteins are generally released to the level of 100–150 mg/L during alcoholic fermentation [2] The mannoproteins from wine *S. cerevisiae* strains contain 0–72% (mostly 10–20%) protein fraction and the remaining polysaccharide fraction generally of 70–98% D-mannose [2–4], which are mainly branched α-(1,2; 1,3) mannans associated with glucans residues [2]. The protein content, mannose compositions, and molecular weights of yeast mannoproteins in wines depend on yeast strains, viticultural practices, and aging conditions [5].

The presence or addition of mannoproteins in wines have been reported to endow diverse advantages, e.g., inhibiting tannin aggregation [4,6]; reducing astringency [7]; keeping phenolic components solubilized in red wines and improving color stabilization [4,8–10]; preventing tartrate crystallization and precipitation in wines [11]; giving wines with rich flavor, body, and mouthfeels [2]; and reducing

protein haze during wine aging [12,13]. Generally, different starters [14] and different commercial mannoproteins from baker's yeasts [4,7] result in wines with different color and tannins contents.

There are two mannoproteins-involved practices to improve wine quality: adding *S. cerevisiae* mannoproteins extracts to wine mashes [4,7,9,10,15] and adding enzymatic preparations to enhance the release of yeast mannoproteins on aging [16]. However, these practices are easy to induce wine off-flavors and spoilage due to uncareful managements [17]. Another strategy is to develop special *S. cerevisiae* mutants with overexpression of mannoproteins by genetic recombination [18] or artificial hybridization of *S. kudriavzevii × S. cerevisiae* [17]. However, gene-modified food microorganisms are subjected to strict regulations in food applications, especially in wines. Wine *S. cerevisiae* mutants with overexpression abilities for high-mannose (M) mannoproteins by food-friendly methods and their applications in winemaking practice are still under investigation.

Accordingly, the purpose of this study was to explore a wine *S. cerevisiae* mutant (noted as CM8) with overexpression ability for high-mannose mannoproteins by food-friendly methods and to characterize the quality attributes of red wines improved by the mutant starter. The multiple benefits and applicability of using the mutant CM8 in winemaking were evaluated, in comparison with those by adding exogenous mannoproteins.

## 2. Materials and Methods

### 2.1. Materials and Chemicals

Kyoho grapes (*Vitis vinifera* L. × *Vitis labracana* Bailey) harvested in winter (January) and summer (July) were obtained from Sinyi Township Farmers Association, Nantou County, Taiwan. The winter (WIN) and summer (SUM) grapes contained sugars at 20.6 and 18.0°Brix, respectively, and similar approximate compositions (Supplementary Table S1). *Saccharomyces cerevisiae* (BCRC 21685)t and *Hansenula mrakii* IFO 0897 were from the Bioresource Collection and Research Center (BCRC) at the Food Industry Research and Development Institution, Hsinchu, Taiwan. Ethyl methane sulfonate (EMS), acetaldehyde, caffeic acid, catechin, gallic acid, glucose, mannose, and sulfur dioxide were from Sigma-Aldrich Chemicals, Inc. (St. Louis, MO, USA). Bovine serum albumin (BSA) was from ThermoFisher Scientific (Waltham, MA, USA). Malvidin-3-glucoside (M3G), quercetin, triethylamine (TEA), sodium dodecyl sulfate (SDS), ferric chloride (FeCl$_3$), yeast extract, peptone, agar, ethanol, isopropanol, and methanol were from Merck (Dermstadt, Germany). Dextrose was from Wako Pure Chemical Industries (Osaka, Japan). The other chemicals used were from Union Chemical Works LTD (Shinchu, Taiwan).

### 2.2. S. Cerevisiae Mutants with Over-Producibility for Mannoproteins

#### 2.2.1. UV Mutagenesis

*S. cerevisiae* was grown in YPD medium (1% yeast extract, 2% peptone and 2% dextrose) at 28 °C for 48 h under shaking at 150 rpm. An aliquot of cell suspension ($1 \times 10^4$ cells/mL, 20 mL) on a 9-cm petri dish was irradiated under a UV lamp of 254 nm [19]. After 35, 45, and 55 s of exposure, 1 mL of the cell suspension was withdrawn for selection of mannoprotein-overexpressed mutants.

#### 2.2.2. Chemical Mutagenesis

Stationary grown cells in the YPD medium were adjusted to about $1 \times 10^4$ cells/mL, i.e., optical density at 600 nm = 1. The culture (5 mL) was subsequently centrifuged at $9,200 \times g$ for 5 min for collecting cell pellet. Cells were dispersed well in 1.7 mL of 0.1 M sodium phosphate buffer (pH 7.0) by vortex. Mutation was conducted by adding 50 μL EMS to the cell suspension and incubating at 30 °C for 15, 30, 45, or 60 min. Reaction was terminated by adding 8 mL of 5% Na$_2$S$_2$O$_3$ [20].

### 2.2.3. Selection of Mannoprotein-Overexpressed Mutants

Killer 9 toxin (k9) agar plate was used to select mannoprotein-overexpressed mutants, which were resistant to k9 due to their cell walls rich in mannoprotein. Killer 9 toxin was produced by culturing *Hansenula mrakii* IFO 0897 (BCRC 23058) in buffered YDP medium (1% yeast extract, 2% peptone, 2% glucose, 50 mM sodium citrate buffer) [21]. After incubated at 19 °C for 14 h with 150 rpm rotation, the resultant k9-containing culture was centrifuged ($9200 \times g$, 10 min) and filtered (0.2 μm sterile filter) to give k9 supernatant. Five mL of k9 supernatant was overlayered on each of buffered YPD agar plates [19]. Serial 10-fold dilutions of each of UV or EMS-treated yeast suspensions (initially $OD_{600}$ = 0.3, ~$10^8$ CFU/mL) were prepared with saline solution. An aliquot of the dilution (100 μL) was sampled to spread on k9-overlayered buffered YPD agar plates. After incubation at 28 °C for 48–72 h, survival colonies were selected. The candidate mutants were selected from k9 toxin plates by the following procedure. A total of 50 colonies were picked from plates for each treatment, 8 colonies among the 50 cultures were selected by second plating, and finally 2 strains (M1 and M7) from UV mutants and 3 strains (CM3, CM5, and CM8) from EMS ones, which showed notable resistances against killer-9 toxin, were further compared their mannoprotein-producing ability.

### 2.2.4. Analysis of Mannose and Glucose Contents in Cell Walls

Survival colonies were enriched on buffered YPD agar plates without k9, followed by washing twice with sterilized miliQ $H_2O$ and collecting by centrifugation at $9200 \times g$ for 10 min at 4 °C. Cells were resuspended in 0.5 mL miliQ $H_2O$ containing 0.5 g of glass beads (diameter of 2 mm) and broken in a MP$^{TM}$ Biomedical FastPrep$^®$-24 Tissue Homogenizer (MP Biomedicals Europe, Eschwege, Germany) by 5 cycles of shakeup and pause for 40 s and 2 min, respectively. Cell debris was collected by centrifugation at $9,200 \times g$ for 10 min at 4 °C, washing with sterile miliQ water trice, and hydrolyzed by 200 μL of 1 M $H_2SO_4$ aqueous solution at ~100 °C (boiling water) for 4 h [3]. The hydrolysate was diluted with 1.8 mL miliQ water, filtered through 0.45 μm nylon membrane, and subjected to a high-performance liquid chromatography (HPLC) system. A Hitachi L-6000 HPLC system equipped with a refractive index detector IOTA 2 (Precision Instruments Inc., Des Plaines, IL, USA) and MetaCarb 87C column ($300 \times 7.8$ mm) (Varian Inc., Palo Alto, CA, USA) was applied. The elution was miliQ water at 0.7 mL/min and 85 °C. The injection volume was 10 μL. Chromatograms were managed with Peak ABC chromatography workstation V 2.11 software. Data were measured in three replications and calibrated with mannose and glucose standard curves.

### 2.2.5. Quantification of Mannoproteins in Yeast Mutants

Mannoproteins in yeast cells were extracted according to Dikit et al. [22]. Yeast cells (dry weight 5 g, approximate 20 g wet weight) were resuspended in 20 mL phosphate buffer (pH 7.0) and autoclaved under 121 °C for 2 h. After cooling and centrifugation ($9,200 \times g$, 10 min), the supernatant was mixed with 5 volumes of 95% ethanol and settled overnight for complete precipitation. The precipitate was washed with 95% ethanol trice and freeze-dried.

Mannoproteins were collected by fast performance liquid chromatography (FPLC). A Pharmacia™ LCC-501 Plus FPLC system equipped with a UV detector (280 nm) and Affinity Chromatography Media Cellufine$^®$ PB resin column (2.6 cm $\times$ 15 cm) (JNC Corporation, Tokyo, Japan) was applied. The column was successively eluted with adsorption buffer (0.01 M sodium phosphate, pH 7.5) for 30 min and, for another 60 min, with eluting buffer (0.01 M sodium phosphate, pH 6.7) containing gradient NaCl from 0.1 M to 0.2 M in 60 min. The flow rate was 5 mL/min. After loading sample into the column, all proteins bound onto resin during running adsorption buffer. Thereafter, mannoproteins were eluted by elution buffer with gradient 0.1–0.2 M NaCl. The eluted mannoproteins were collected, dialyzed (molecular weight cut off = 1 kDa) against deionized water to remove salts, precipitated with 5 volumes of 95% ethanol, freeze-dried, and weighted. The yield of mannoproteins was examined in three replications and on dry yeast basis.

### 2.3. Production of Red Wine

Kyoho grapes were washed, mechanically crashed, and pasteurized (70 °C, 10 min). Three batches of red wine were produced. For each batch, approximate 3.5 kg pasteurized grapes including pulp, peel and grapes juice (1.0 kg), which recovered from grapes crashing processes, were mixed with brown sugar (to be final 25 °Brix) in a 5-liter glass jar and inoculated with yeasts (500 mL, $1 \times 10^8$ CFU/mL). The mixture was pre-fermented at 18 °C for 6 h and performed primary fermentation at 28 °C for 12 d. After primary fermentation and removing the residues, wine mash was subjected to secondary fermentation at 4 °C for 14 d. After removing precipitates, the given red wine was transferred to a sterile glass jar (5 L) and incubated at 4 °C for 90 d to complete the aging process.

### 2.4. Physicochemical Determinations for Red Wines

### 2.4.1. pH Value

A pH meter (EL20, Mettler Toledo, Schweiz) was employed to determine the pH value of wine.

### 2.4.2. Ethanol Concentration

Wine sample (10 mL) was diluted by 500 folds with deionized water and filtered through 0.45 μm nylon membrane before gas chromatography. An Agilent™ 7890A GC system (Agilent™ Inc., Palo Alto, CA, USA) equipped with an FID detector and Rtx-1301 column (30 m × 0.32 mm ID × 1.0 μm) (RESTK Corporation, Bellefonte, PA, USA) was employed. Helium was used as the carrier gas. The temperature program started from 35 °C (hold for 5 min) to 155 °C at a heating rate of 20 °C/min for column oven. Temperatures for injector and detector were set at 200 and 260 °C, respectively. The linear velocity of carrier gas was 25 cm/sec at 35 °C and a split ratio of 30:1. Isopropanol (5% *v/v*), as the internal standard, was added into the wine sample. One μL of sample was applied to the GC. Data were measured in three replications.

### 2.4.3. Titratable Acidity

According to the method of OIV [23], a portion of wine sample (1 mL, de-$CO_2$) was diluted with 10 mL de-$CO_2$ distilled water and titrated with 0.10 N NaOH aqueous solution (equivalent factor F = 0.075) to an end point of pH 8.1. The volume of NaOH solution consumed ($V_{NaOH}$, mL) was record. The titratable acidity was calculated using the following equation, Equation (1), where $N_{NaOH}$ = the normality of NaOH solution (0.10); D = dilution fold (10), $V_s$ = sample volume (1 mL), and expressed as gram of tartaric acid equivalent (TAE) per liter of wine. Data were measured in three replications.

$$\text{Titratable acidity (g/100 mL)} = 100 \times V_{NaOH} \times N_{NaOH} \times F \times D/Vs. \tag{1}$$

### 2.4.4. Total Phenolics, Flavonoids, and Tartaric Ester Contents

According to Cliff et al. [5], an aliquot of sample (0.5 mL) was diluted with 5 mL of 10% ethanol. The diluted sample was sampled (0.25 mL) and added to 0.25 mL of 0.1% HCl in 95% ethanol and 4.55 mL of 2% HCl. Each mixture was vortexed and allowed to stand for 15 min before measurement on the absorbances at 280, 320, and 360 nm for corresponding concentration of total phenolics, flavonols, and tartrate esters, respectively. Based on standard calibration curves, the total phenolics content (TPC) was expressed as mg gallic acid equivalent (GAE) per liter of wine; total flavonols content (TFC), as mg quercetin equivalent (QE) per liter of wine; and total tartrate esters content (TEC), mg caffeic acid equivalent (CAE) per liter of wine. Data were measured in three replications.

### 2.4.5. Anthocyanins Composition Assay

Anthocyanins content expressed as malvidin-3-glucoside (M3G) ($AC_{M3G}$) was measured with a HPLC system equipped with GRACE Alltima C18 column (5 μm diameter, Fisher Scientific, Osaka, Japan). The mobile phase was a linear gradient of solvents A (5% formic acid) and B (methanol containing 5% formic acid) at a flow rate of 0.8 mL/min. The proportion of solvent B was programmed as follows: 0–2 min, 0–15%; 2–31 min, 15–45%; 31–32 min, 45–15%; and 32–45 min, 15%. The injection volume was 10 μL. The elution was monitored at 520 nm. Based on malvidin-3-glucoside standard curve, data were measured in three replications and presented as mg of M3G per liter of wine.

Besides, different types of anthocyanins (AC), including co-pigmented, monomeric, and polymeric ACs, were determined according to their reactivities with acetaldehyde and $SO_2$ [5]. A portion of wine (2 mL) was successively added with 20 μL of 20% acetaldehyde, mixed well, stayed for 45 min, and measured for the absorbance at 520 nm (noted as $A^{ace}$) in a Spectrophotometer U-1800 system (Hitachi, Tokyo, Japan). Similar procedure was done on another portion of wine (2 mL) added with 160 μL of 5% ($w/v$) $SO_2$, giving the absorbance at 520 nm ($A^{SO2}$). A portion of wine sample without acetaldehyde or $SO_2$ was directly measured the absorbance at 520 nm ($A^{wine}$). The concentrations of total, co-pigmented, polymeric, and monomeric AC were positively related to the values of $A^{ace}$, ($A^{ace}-A^{wine}$), $A^{SO2}$, and ($A^{wine}-A^{SO2}$), respectively [5]. Because that the absorbance is fundamentally proportional to the AC concentration, the percentages of different types of ACs in total AC can be calculated as Equations (2)–(4). All data were measured in three replications.

$$\text{Co-pigmented AC (\%)} = 100 \times (A^{ace}-A^{wine})/A^{ace}, \tag{2}$$

$$\text{Polymeric AC (\%)} = 100 \times A^{SO2}/A^{ace}, \tag{3}$$

$$\text{Monomeric AC (\%)} = 100 \times (A^{wine}-A^{SO2})/A^{ace}. \tag{4}$$

### 2.4.6. Total Tannin Content

Total tannin content (*TTC*) in wine was determined according to the method of Harbertson et al. [24]. Briefly, tannins in wine were tested based on a successive procedure including precipitation with bovine serum albumin at pH 4.9, centrifugation to give tannin-BSA pellet, dissolution in TEA buffer containing 5% TEA ($v/v$) and 10% SDS ($w/v$), and reaction with ferric chloride reagent (10 mM $FeCl_3$ in 0.01 N HCl). The product was measured for the absorbance at 510 nm. Catechin was used as the standard. Data were measured in three replications and expressed as mg of catechin equivalent (CE) per liter of wine.

### 2.4.7. Chromatic Characteristics

For CIE colorimetry, color parameters (*L* \*, *a* \*, and *b* \*) were recorded in a ColorQuest HunterLab spectrocolorimeter equipped with HunterLab Universal Software Version 3.0 (HunterLab, Hunter Associates Laboratories Inc., Reston, VA, USA). Color difference (Δ*E*) between two samples was calculated according to Equation (5):

$$\Delta E^2 = (L *_1 - L *_2)^2 + (a *_1 - a *_2)^2 + (b *_1 - b *_2)^2 \tag{5}$$

For spectrometry, wine sample was filtered through 0.45 μm membranes and detected for the absorbances at 420, 520, and 700 nm in a 0.2 cm path-length quartz cuvette in a Spectrophotometer U-1800 system (Hitachi, Tokyo, Japan). The color density and hue were calculated as Equations (6) and (7) [5], where $A_{420}$, $A_{520}$, and $A_{700}$ were the absorbances at 420, 520, and 700 nm, respectively.

$$\text{Color density} = [(A_{520} - A_{700}) + (A_{420} - A_{700})], \tag{6}$$

$$\text{Color hue} = [(A_{420} - A_{700})/(A_{520} - A_{700})]. \tag{7}$$

### 2.5. Sensory Evaluation of Red Wine

Consumer acceptance and preference tests were performed on four kinds of wines made of winter (WIN) or summer (SUM) grapes fermented with parent strain (*S. cerevisiae BCRC 21685*, SC) or mutant CM8. A total of 73 consumer-type panelists (22 males and 51 females) included staffs and students from the Department of Food and Nutrition, Providence University (Taiwan). All tested samples were adjusted to 3 °Brix, according to the result of consumer preference pretest that slightly sweetened wine was more popular than blank wine, and kept in a 4 °C refrigerator before sensory evaluation. Consumer acceptance test was done by 9-point hedonic test with the following standards. The consumer acceptance index was calculated as 100 × means of given points/9.

Like extremely: 9 points　　　　　　　　　　Dislike slightly: 4 points
Like very much: 8 points　　　　　　　　　　Dislike moderately: 3 points
Like moderately: 7 points　　　　　　　　　　Dislike very much: 2 points
Like slightly: 6 points　　　　　　　　　　　Dislike extremely: 1 point
Neither like nor dislike: 5 points

Consumer preference test was done with Rank preference, ISO standards. The evaluated characteristics of wine samples included color, aroma, flavor, acidity, sweetness, astringency, and overall preference.

### 2.6. Statistical Analysis

For chemical compositions, analysis of variance was performed by one-way analysis of variance (one-way ANOVA) procedures using SPSS 9.3 software (SPSS Inc. Chicago, IL, USA). Duncan's new multiple-range test was used to determine the differences among means. A significance level at $p < 5\%$ was adopted for all comparisons. XLSTAT software (Addinsoft Inc., New York, NY, USA) was applied for the results of sensory evaluation. ANOVA and Honestly Significant Difference analysis were conducted for consumer acceptance data (9-point hedonic data and acceptance index). Principal component analysis (PCA) was performed on the characteristics of red wines. And, Friedman test for Rank preference, ISO standard was conducted on consumer preference results.

## 3. Results

### 3.1. Mannoproteins Contents of S. Cerevisiae Mutants

Colonies, grew in killer-9 toxin containing YPD Figure 1 shows the colonies of parent strain, *S. cerevisiae BCRC* 21685 (SC), and its mutant strains grown on k9-overlayered agar plate, where mutants M1 and M7 were induced by UV light; and CM3, CM5 and CM8 were by EMS mutagenesis. Generally, five mutants (M1, M7, CM3, CM5, and CM8) grew well at $10^{-2} - 10^{-4}$ dilutions (D2-D4) and were apparently more tolerant toward killer-9 toxin than SC, which grew badly at $10^{-2}$ dilution (D2). Accordingly, these mutants were further examined for the monosaccharide compositions and mannoproteins contents in cell wall.

Table 1 displays that the cell wall of SC possessed, in average, 151 mg/g glucose (G) and 298 mg/g mannose (M), giving an averaged M/G molar ratio = 1.97. Those of mutants M1, M7, CM3, and CM5 displayed generally 98–126 mg/g G, 215–279 mg/g M, and a M/G molar ratio in the range of 2.06–2.37, not significantly ($p > 0.05$) different from the SC. Interestingly, mutant CM8 cell wall had in average 111.9 mg/g G, 477.6 mg/g M, leading to a M/G molar ratio of 4.27, significantly ($p < 0.05$) higher than the SC and the other mutants. That is, mannose in the carbohydrate moiety of mannoproteins accounted for 81% for CM8, in contrast to SC (66%). The mannoprotein content of CM8 cell wall was in average 386.8 mg/g, significantly ($p < 0.05$) greater than the SC (340 mg/g). Accordingly, CM8 mutant was used as a starter to produce red wine, in comparison with SC.

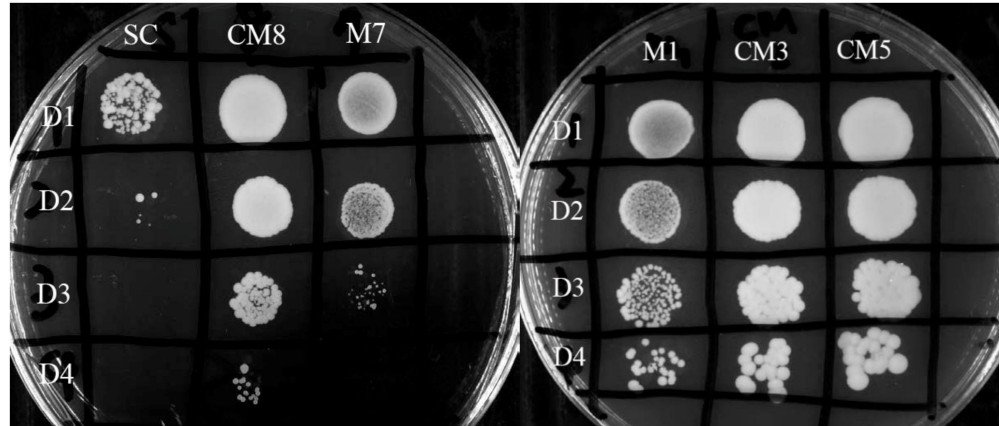

**Figure 1.** Images on yeast colonies on YPD citrate agar plates containing killer-9 toxin. SC = parent strain, *S. cerevisiae BCRC* 21685; M1 and M7 = mutants by UV light; CM3, CM5, and CM8 = mutants by ethyl methane sulfonate (EMS); D1 to D4 = dilutions of $10^{-1}$ to $10^{-4}$ for yeast suspensions (initially ~$10^8$).

**Table 1.** Sugar compositions and mannoprotein contents in the cell walls of *S. cerevisiae* and its mutant strains.

| Strain [A] | Glucose (G) (mg/g) | Mannose (M) (mg/g) | M/G molar ratio | Mannoproteins (mg/g) |
|---|---|---|---|---|
| SC | 151.0 ± 1.3 [a,B] | 298.0 ± 15.7 [b] | 1.97 ± 0.09 [b] | 340.1 ± 0.6 [b] |
| M1 | 97.5 ±8.6 [c] | 231.2 ± 20.3 [cd] | 2.37 ± 0.00 [b] | – [C] |
| M7 | 104.2 ±2.1 [bc] | 215.2 ± 6.5 [d] | 2.06 ±0.02 [b] | – |
| CM3 | 126.1 ± 2.7 [b] | 267.0 ± 6.7 [bc] | 2.12 ±0.10 [b] | – |
| CM5 | 123.5 ± 11.6 [b] | 278.5 ± 28.3 [b] | 2.26 ±0.01 [b] | – |
| CM8 | 111.9 ± 10.2 [b] | 477.6 ± 12.5 [a] | 4.27±0.26 [a] | 386.8 ± 0.3 [a] |

[A] SC: *S. cerevisiae BCRC*21685; M1 and M7: mutants induced by UV light; CM3, CM5, and CM8: mutants induced by ethyl methane sulfonate (EMS). [B] Data are presented as means ± S.D. (n = 3); data followed with different superscripts (a–d) differ significantly ($p < 0.05$) in the same column. [C] Not determined.

### 3.2. Quality of Red Wine Fermented by S. Cerevisiae Mutants

Table 2 shows the characteristics of four different red wines fermented by parent strain (SC) or mutant (CM8) on winter grapes (WIN) or summer grapes (SUM). Evidently, all wines studied showed similar pH values (3.93–3.94) and alcohol concentrations (in average 117 g/L for SC and 123 g/L for CM8). The titratable acidity was significantly ($p < 0.05$) greater for SC wines (1.18–1.23 g TAE/L) than for CM8 wines (1.01–1.09 g TAE/L). The wines made from winter grapes (SC-WIN and CM8-WIN) contained significantly ($p < 0.05$) higher contents of total phenolics (*TPC*), tartaric esters (*TEC*), total flavonols (*TFC*), anthocyanins (indicated by $AC_{M3G}$ and $A^{ace}$), and total tannins (*TTC*), and appeared with significantly ($p < 0.05$) greater *a* *, *b* *, and color density together with lessened *L* * and color hue, in contrast to their SUM counterparts. Focusing on the WIN wines, both SC-WIN and CM8-WIN exhibited very similar *TPC* (1409–1439 mg GAE/L), *TEC* (774–784 mg CAE/L), *TFC* (255–283 mg QE/L), $AC_{M3G}$ (11.01–11.17 mg M3G/L), $A^{ace}$ (1.436–1.484), and color hue (1.09–1.10). In difference, CM8-WIN showed significantly ($p <0.05$) greater total anthocyanins ($A^{ace}$), *TTC*, *a* *, *b* *, and color density, than those did SC-WIN. For anthocyanins types, the co-pigmented, monomeric, and polymeric ones accounted for 0.8–3.1%, 35.5–37.4%, and 61.0–62.5%, respectively, for all wines. Basically, $A^{ace}$ value was found very closely related to $AC_{M3G}$: $A^{ace} = 0.139 \times AC_{M3G} - 0.0855$ ($R^2 = 0.995$). Besides, the color differences between WIN and SUM wines ($\Delta E_{WIN-SUM}$) appeared to be 28.73 and 29.72 for SC and CM8, respectively, significantly greater than those between SC and CM8 samples ($\Delta E_{SC-CM8} = 1.64$ and 0.63 for WIN and SUM, respectively) (data not tabulated).

**Table 2.** Effects of starters and seasonal grapes on the chemical and colorimetric characteristics of the red wines studied [A].

| Parameter | Unit [B] | SC-WIN | SC-SUM | CM8-WIN | CM8-SUM |
|---|---|---|---|---|---|
| pH | | 3.94 | 3.93 | 3.94 | 3.94 |
| Alcohol | (% vol) | 11.90 ± 0.0 [b,C] | 12.00 ± 0.0 [b] | 12.56 ± 0.0 [a] | 12.52 ± 0.0 [a] |
| Titratable acidity | (g TAE/L) | 1.23 ± 0.04 [a] | 1.18 ± 0.04 [a] | 1.01 ± 0.00 [b] | 1.09 ± 0.04 [b] |
| Total phenolics (TPC) | (mg GAE/L) | 1409 ± 1 [a] | 1034 ± 62 [b] | 1439 ± 37 [a] | 962 ± 21 [c] |
| Tartaric esters (TEC) | (mg CAE/L) | 784 ± 18 [a] | 558 ± 40 [b] | 774 ± 21 [a] | 517 ± 7 [c] |
| Total flavonols (TFC) | (mg QE/L) | 255 ± 52 [ab] | 206 ± 1 [b] | 283 ± 4 [a] | 205 ± 6 [b] |
| Anthocyanins ($AC_{M3G}$) | (mg M3G/L) | 11.17 ± 0.03 [a] | 5.21 ± 0.30 [b] | 11.01 ± 0.06 [a] | 5.63 ± 0.23 [b] |
| Total anthocyanins | $A^{ace}$ | 1.436 ± 0.002 [b] | 0.659 ± 0.001 [d] | 1.484 ± 0.003 [a] | 0.677 ± 0.002 [c] |
| Co-pigmented | % | 2.0 ± 0.2 [c] | 2.9 ± 0.2 [ab] | 3.1±0.0 [a] | 0.8 ± 0.1 [d] |
| Monomeric | % | 37.4 ± 0.3 [a] | 35.5 ± 0.5 [c] | 35.9 ± 0.1 [c] | 36.7 ± 0.2 [b] |
| Polymeric | % | 60.6 ± 0.1 [d] | 61.6 ± 0.3 [b] | 61.0 ± 0.0 [c] | 62.5 ± 0.1 [a] |
| Total tannins (TTC) | (mg CE/L) | 310.0 ± 8.8 [b] | 244.4 ± 11.5 [d] | 405.0 ± 1.8 [a] | 274.4 ± 6.2 [c] |
| L * | | 55.50 ± 0.01 [c] | 76.85 ± 0.07 [a] | 54.07 ± 0.00 [d] | 76.28 ± 0.00 [b] |
| a * | | 38.48 ± 0.00 [b] | 19.60 ± 0.01 [d] | 39.12 ± 0.01 [a] | 19.75 ± 0.00 [c] |
| b * | | 13.02 ± 0.01 [b] | 9.43 ± 0.00 [d] | 13.52 ± 0.00 [a] | 9.66v0.01 [c] |
| Color density | | 2.269 ± 0.004 [b] | 1.139 ± 0.000 [d] | 2.321 ± 0.006 [a] | 1.174 ± 0.001 [c] |
| Color hue | | 1.091 ± 0.001 [c] | 1.320 ± 0.007 [a] | 1.100 ± 0.002 [c] | 1.295 ± 0.006 [b] |
| $\Delta E_{WIN-SUM}$ [D] | | 28.73 | | 29.72 | |

[A] CM8 = mutant of S. cerevisiae BCRC 21685; SC = parent strain, S. cerevisiae BCRC 21685; SUM and WIN: Kyoho grapes harvested in summer and winter seasons, respectively. [B] CAE = caffeic acid equivalent; CE = catechin equivalent; GAE = gallic acid equivalent; M3G = malvidin-3-glucoside; QE = quercetin equivalent; TAE = tartaric acid equivalent. [C] Data are presented as means ± S.D. (n = 3); data followed with different superscript letters (a–d) differ significantly ($p < 0.05$) in the same row. [D] Color difference between wines made of WIN and SUM.

## 3.3. Sensory Properties of Red Wine Fermented by S. Cerevisiae Mutants

Figure 2A shows the results of consumer acceptance test for four wines. It is indicated that, for color, both WIN wines (CM8-WIN and SC-WIN) got generally more panelists for high preference scores, in contrast to those did SUM wines (CM8-SUM and SC-SUM). For aroma (Figure 2B), all samples showed generally similar preference profiles. For flavor (Figure 2C), both CM8 wines (CM8-SUM and CM8-WIN) were evaluated with high scores by more panelists than their SC counterparts. And, for overall preference (Figure 2D), both CM8 wines got more panelists for high scores than did SC wines, in which the sum of panelists scored 7–9 for CM8-SUM, CM8-WIN, SC-SUM, and SC-WIN were 13, 21, 11, and 16, respectively.

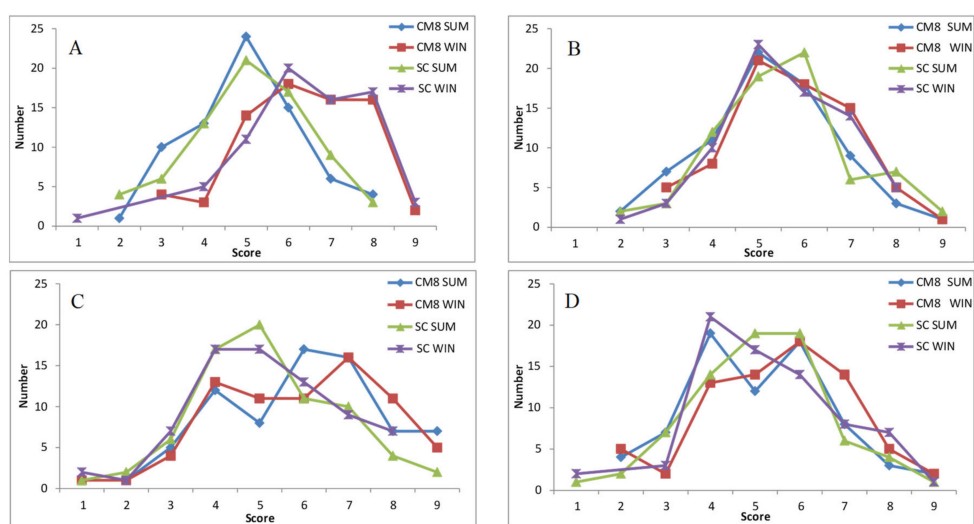

**Figure 2.** Sensory evaluation results on color (**A**), aroma (**B**), flavor (**C**), and overall preference (**D**) of four red wines. *Y* axis = panelist number; *X* = hedonic 9-ponit score. Samples are explained in Table 2.

Table 3 shows the consumer acceptance and rank preference for the wines studied. For color, both WIN wines obtained significantly ($p < 0.05$) higher preference scores than both SUM wines. For flavor, both CM8 wines got scores significantly greater than both SC wines. For aroma and overall preference, all wines get scores insignificantly different between each other. The resultant acceptance index (AI) was in the order of CM8-WIN (65) > SC-WIN (62) > CM8-SUM (59) > SC-SUM (58). The sum of preference rank by Friedman test was the lowest, i.e., most preferred, for CM8-WIN, followed by SC-WIN, CM8-SUM, and SC-SUM, agreeing with AI. Generally, the wines showed similar degrees of sweetness, acidity, and astringency. Conclusively, CM8-WIN, the topped one rich in total phenolics, tartaric esters, total anthocyanins, and tannins contents, displayed the greatest colorimetric a *, b *, color density, and highest consumer acceptance in color, flavor, aroma, overall, *AI*, and consumer preference among the wines examined.

**Table 3.** Consumer acceptance and rank preference data for the red wines studied [A].

| Parameter | SC-WIN | SC-SUM | CM8-WIN | CM8-SUM |
|---|---|---|---|---|
| Color | 6.45 ± 1.45 [a,B] | 5.10 ± 1.45 [b] | 6.30 ± 1.46 [a] | 5.04 ± 1.38 [b] |
| Flavor | 5.14 ± 1.65 [b] | 5.16 ± 1.63 [b] | 5.93 ± 1.84 [a] | 6.00 ± 1.76 [a] |
| Aroma | 5.56 ± 1.32 [a] | 5.53 ± 1.50 [a] | 5.67 ± 1.36 [a] | 5.25 ± 1.44 [a] |
| Overall preference | 5.27 ± 1.60 [a] | 5.12 ± 1.54 [a] | 5.51 ± 1.66 [a] | 5.08 ± 1.63 [a] |
| *AI* [C] | 62 | 58 | 65 | 59 |
| Sum of ranks [D] | 177 (2.43 [ab]) | 201 (2.75 [b]) | 159 (2.18 [a]) | 194 (2.65 [ab]) |
| Degree of sweetness | 2.32 ± 1.01 [a] | 2.10 ± 1.16 [a] | 2.34 ± 1.25 [a] | 2.43 ± 1.03 [a] |
| Degree of acidity | 3.33 ± 1.53 [a] | 3.14 ± 1.31 [a] | 3.33 ± 1.25 [a] | 3.30 ± 1.48 [a] |
| Degree of astringency | 3.16 ± 1.32 [a] | 3.11 ± 1.24 [a] | 3.14 ± 1.02 [a] | 2.86 ± 1.26 [a] |

[A] Samples are explained in Table 2. [B] Data are presented as means ± S.D. ($n = 73$); data followed with different superscript letters (a–b) differ significantly ($p < 0.05$) in the same row. [C] *AI* = acceptance index =100 *(averaged scores/9) based on Hedonic 9-point tests for color, flavor, aroma, and overall preference. [D] By Friedman test, the lower the sum of ranks, the higher was the preference by the consumers. Data in parenthesis are means of ranks ($n = 73$).

### 3.4. Principal Factors Governing Sensory Preference of Wines

Figure 3 depicts the PCA results on the chemical, colorimetric, and some sensory data of the red wines studied. It is evident that factors F1 and F2 explained 92.67% of data variations, where F1 accounted for 75.85%; and F2 did for 16.82%. Active variables, such as degree of sweetness, pH, degree of acidity, various phenolic compositions (total phenolics, tartaric esters, flavonols, anthocyanins, and tannins), and degree of astringency, contributed positively to F1, where degree of astringency did negatively to F2. Active variables *L* * value, color hue, and titratable acidity contributed negatively to F1, where titratable acidity did negatively to F2. Briefly, F1 concerned mainly color and phenolic compositions (including total phenolics, flavonols, tartaric esters, anthocyanins, and tannins) and was positive for high color (indicated by high color density, *a* *, and *b* *) and high phenolic compositions. F2 concerned mainly flavor and titratable acidity and positive for high flavor and low titratable acidity. Quadrant I with CM8-WIN was characterized with high color, high phenolics, and high flavor. Quadrant II with CM8-SUM was with low color, low phenolics, and high flavor. Quadrant III with SC-SUM was with low color, low phenolics, and low flavor. And, quadrant IV with SC-WIN was with high color, high phenolics, and low flavor. Besides, CM8-SUM in quadrant II, in the opposite side of quadrant IV with vector degree of astringency, showed low astringency.

Interestingly, the PCA biplot (Figure 3) illustrates that the vectors of phenolic compositions (total anthocyanins, flavonols, anthocyanins, total phenolics, tartaric esters, and tannins) and CIE parameters (a *, b *, and color density) were very close, implying that these parameters were closely correlated to each other. Total phenolic content (TPC) could be a key index for describing the variations in the above phenolic compositions and CIE parameters. The results are shown in Table 4. The relationships are: total tartaric esters (TEC), $0.565 \times$ TPC $- 25.65$ (correlation coefficient $R^2 = 0.994$); total flavonols (TFC), $0.149 \times$ TPC $+ 56.7$ ($R^2 = 0.927$); anthocyanins as M3G ($AC_{M3G}$), $0.0130 \times$ TPC $- 7.48$ ($R^2 = 0.966$);

and total tannins (TTC), 0.229 × TPC + 31.16 ($R^2$ = 0.662). All colorimetric parameters and sensory color could be expressed by TPC (in g GAE/L) as follows: L * = −50.3 × TPC + 127 ($R^2$ = 0.981); a * = 44.2 × TPC − 24.3 ($R^2$ = 0.984); b * = 8.60 × TPC + 0.998 ($R^2$ = 0.972); color density = 2.63 × TPC − 1.45 ($R^2$ = 0.980); color hue = −0.484 × TPC + 1.79 ($R^2$ = 0.954); and sensory attribute color = 3.01 × TPC + 2.07 ($R^2$ = 0.976). Generally, these functions of TPC described 92.7–99.4% of the data variations except TTC, which was accounted by TPC for only 66.2%.

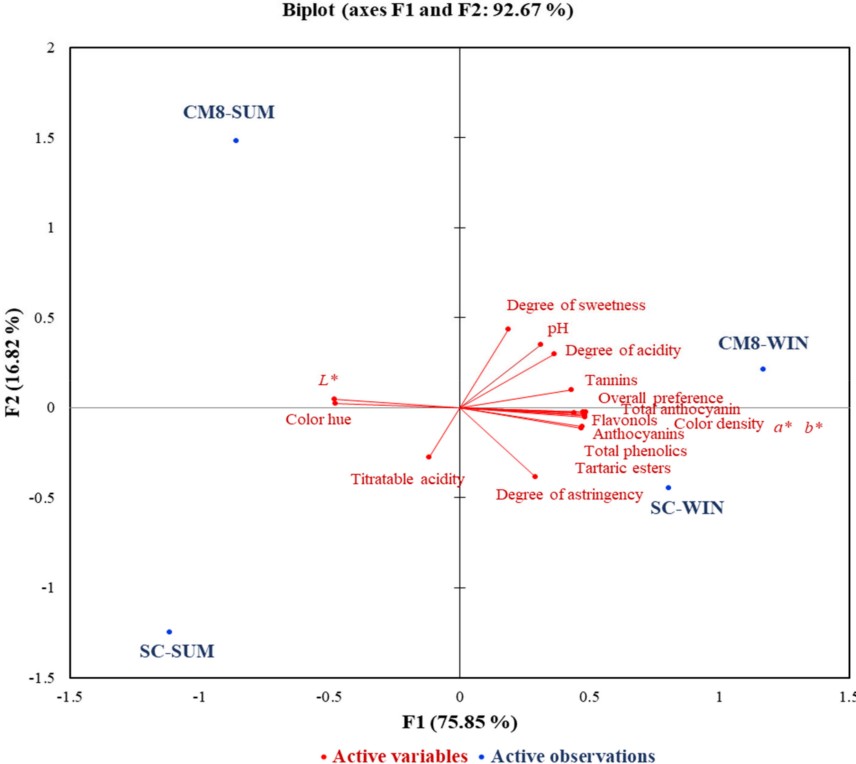

**Figure 3.** Principal component analysis (PCA) biplots for chemical compositions, CIE colorimetric parameters, and sensory attributes of four red wines. Samples are explained in Table 2.

**Table 4.** Changes in various phenolic compositions, colorimetric parameters, and sensory color score of four red wines studies with respect to total phenolics content (*TPC*).

| Variable | Function | $R^{2\ C}$ |
|---|---|---|
| Total tartaric esters content (*TEC*) | 0.565 × *TPC* − 25.65 [A] | 0.994 |
| Total flavonols content (*TFC*) | 0.149 × *TPC* + 56.7 [A] | 0.927 |
| Anthocyanins content ($AC_{M3G}$) | 0.0130 × *TPC* − 7.48 [A] | 0.966 |
| Total tannins content (*TTC*) | 0.229 × *TPC* + 31.16 [A] | 0.662 |
| L * | −50.3 × *TPC* + 127 [B] | 0.981 |
| a * | 44.2 × *TPC* − 24.3 [B] | 0.984 |
| b * | 8.60 × *TPC* + 0.998 [B] | 0.972 |
| Color density | 2.63 × *TPC* − 1.45 [B] | 0.980 |
| Color hue | −0.484 × *TPC* + 1.79 [B] | 0.954 |
| Sensory color score | 3.01 × *TPC* + 2.07 [B] | 0.976 |

[A] *TPC* in mg GAE/L. [B] *TPC* in g GAE/L. [C] Correlation coefficients by the least-squared fits.

## 4. Discussion

Evidently, the mannose content in SC mannoproteins was 66% and increased to 81% for that in CM8 mannoproteins. The mannose content in mannoproteins from SC, a wine *S. cerevisiae* strain, is clearly greater than those from baker's *S. cerevisiae* strains (39–55%) [25,26]. The mannose content can

be increased to 93–97% by isolation assisted with β-(1,3)-glucanase action [25]; however, enzymatic treatments are expansive for industry.

Comparatively, the WIN wines made from Kyoho grape show greater levels of total phenolics, tartaric esters and flavonols, lower titratable acidity, notably lesser anthocyanins level, and comparable tannins content range, in contrast to 78 commercial red wines [5], which exhibit generally pH 3.66–3.72; titratable acidity = 5.80–6.24 g TAE/L; *TPC* = 932–1063 mg GAE/L; *TEC* = 128–162 mg CAE/L; *TFC* = 39–64 mg QE/L; $AC_{M3G}$ = 61–125 mg M3G/L; and *TTC* = 266–414 mg CE/L.

There are some similarities and differences between the PCA biplots for the wines studied and 173 commercial wines from British Columbia in Canada [5]. In similarity, all phenolic compositions examined, CIE *a* \*, *b* \* and color density, and sensory attribute astringency contribute positively to factor F1. While *L*\* and color hue contribute negatively to F1 but positively to F2. Quadrant I is characterized with high phenolics and high tannins; quadrant II, low color (and low phenolics); quadrant III, low phenolics (and low color); and quadrant IV, high color and high anthocyanins. In difference, in this study vector astringency was in quadrant IV, away from vector tannins (quadrant I), pointing to that these two variables are not closely correlated to each other, different from the cases of regular wines that astringency is mostly dependent on tannins level [5,24]. Vectors total phenolics, tartaric esters, flavonols, anthocyanins, and color density, *a* \*, and *b* \* were very close to each other and located in the center between quadrants I and IV, i.e., contributed almost equally to both quadrants concerning CM8-WIN and SC-WIN. Vector titratable acidity was in quadrant III, not in quadrant I as reported for commercial wines [5]. All these differences could be reasonably linked to the effects of CM8 high-M mannoproteins by interactions with wine compositions (tannins, flavonols, anthocyanins, tartaric acid, and aroma), especially tannins, in addition to the effects of grape compositions.

Focusing on the mannoproteins effects, the promising effects of enriched high-M mannoproteins from CM8 could be responsible for the high-quality attributes (color, flavor, and consumer acceptance and preference) of CM8-WIN wines, which possessed greater *TTC* and slightly higher $A^{ace}$ (total anthocyanins), in contrast to SC-WIN. These effects are similar to those by addition of yeast mannoproteins on reducing astringency, improving sensory characteristics, and increasing aroma persistence for red wines [4,7,27], stabilizing anthocyanins-dominated color, and improving the taste of blueberry wine [10]. Mannoproteins-tannins interactions seem to be noticeable in this study, agreeing with the binding abilities of baker's yeast mannoproteins that are stronger for tannins than the other phenolic compounds [26]. However, the potential interactions of mannoproteins with the other wine compositions, such as proanthocyanins [4,6], flavonols [7], tartaric acid [11], and aroma compounds [2], may not be neglected. These interactions facilitate the solubilization, rather than precipitation, of the related compositions in wine. This is further confirmed by the observation that CM8 wine mashes showed half-volume precipitates of that SC in the end of fermentation (data not shown).

Briefly, using a wine starter of higher-M mannoproteins (>80% mannose), like *S. cerevisiae* CM8, is more feasible than those of lower-M mannoproteins, such as SC (66% mannose), or baker's yeasts (50–70% mannose) to improve the overall quality attributes of red wines. CM8 mutant may be a good starter competitive and beneficial for wine industry.

## 5. Conclusions

This study has discovered a new *S. cerevisiae* mutant CM8 with overexpressed high-M mannoproteins. Using CM8 or parent *S. cerevisiae* (SC) as starters for alcoholic fermentation on winter Kyoho grapes (WIN), the resultant wines appeared to be of significantly greater total tannins and total anthocyanins levels, higher color, and higher flavor for CM8-WIN than for SC-WIN wines. By PCA biplots, CM8-WIN wine was classified in quadrant I characterized with high color, high phenolics, and high flavor; and SC-WIN in quadrant IV, high color, high phenolics, low flavor, and high astringency. The improvements in these wine characteristics by CM8, in contrast to SC, could be attributed to its high-M mannoproteins via the interactions with related wine compositions, resembling those by addition of exogenous yeast mannoproteins reported. Conclusively, using CM8 as a starter to

ferment Kyoho grapes is feasible to improve wine quality in situ during fermentation and competitive to the effects of adding exogenous mannoproteins in winemaking. More investigations on the effects of CM8 mutant or other yeast strains with various mannose contents in mannoproteins on the quality attributes of diverse wines are needed in the future.

**Supplementary Materials:** The following are available online at http://www.mdpi.com/2227-9717/8/11/1483/s1.

**Author Contributions:** Conceptualization, Y.-C.C.; funding acquisition, Y.-C.C.; writing, P.F.-H.L. and Y.-C.C.; formal analysis, P.-C.H.; data curation, P.-C.H. and B.-K.L.; investigation, Y.-C.C., P.L., R.D.D. and P.-M.W.; methodology, B.-K.L. and P.-M.W. All authors have read and agreed to the published version of the manuscript.

**Funding:** This work was supported by the Ministry of Science and Technology, R.O.C. (MOST 104-2320-B2-126-004-MY3).

**Conflicts of Interest:** The authors have declared that they have no conflict of interest with respect to the work described in this manuscript.

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
