# Peer review of "Improved Phenolic Compositions and Sensory Attributes of Red Wines by Saccharomyces cerevisiae Mutant CM8 Overproducing Cell-Wall Mannoproteins"

_processes, doi:10.3390/pr8111483_

Round 1

Reviewer 1 Report

The article describes the isolation of a mannoprotein overproducing mutant of Saccharomyces cerevisiae by a method previously described by other authors. The mutant has been used in the production of red wine and the results have been compared with those of a strain producing normal levels of mannoproteins. The introduction provide sufficient background on the subject. The study is well planned, and has been carried out correctly using known and reliable methodologies.
The results are described correctly and orderly, including figures and tables, and the conclusions are consistent with the results.

Minor Points:

Title: all the mannoproteins produced by S. cerevisiae ara High mannose type.

I suggest:

Improved phenolic compositions and sensory attributes of red wines by Saccharomyces cerevisiae mutant CM8 overproducing cell-wall mannoproteins

Spelling

Line186: says: tannins in wine "was" tested    

Should say: "were"

Line 193, says: "By" CIE clorimetry

Should say: "For" CIE colorimetry

Line 197 says: "By" spectrometry

Should say: "For" spectrometry

Line 204 says: "Totally" 73 consumer-type

Should say: "A total of" 73 consumer-type

Reviewer 2 Report

Dear editor and authors,

In the presented study, a Saccharomyces cerevisiae strain underwent mutagenesis and was selected for high production of mannoproteins. Mannoprotein production was analyzed quantitatively and qualitatively. Subsequently, red wine was produced with a selected mutated strain and the control strain. Evaluation demonstrated that wine fermented with the mutated and selected strain showed higher content in excreted mannoproteins, and was attributed more positive characteristics analytically and by a sensory panel.

The study is well conceptualized and performed. It builds on the reproduction of similar existing studies (e.g., González-Ramos et al., 2010; Quirós et al., 2010). The main novelty of this study is a more thorough analysis of the mutated strains’ mannoprotein fraction, the assessment of additional wine metabolites and the evaluation of produced wines by a sensory panel. In addition, a new improved S. cerevisiae strain is provided for commercial wine production. In summary, the findings are of general interest for researchers and producers engaged in (red) wine production.

General remarks:

The mutation of S. cerevisiae and subsequent selection of CM8 is an important part of the study. However, it is neither represented in the abstract nor the introduction. Now, the impression is given that CM8 was a preexisting strain. Please update the publication accordingly.

In the Bioresource Collection and Research Center, Saccharomyces cerevisiae strain BRCR21685 is labelled as rice wine strain. Please critically reflect if this could have influenced obtained results (e.g., could a wine strain have showed higher mannoprotein excretion from the beginning, therefore limiting the potential of optimization through mutation) and state why a more common grape wine strain like EC1118 was not used. If necessary, include this reflection in the publication.

Several times throughout the manuscript, results that were previously shown to be not significant are stated to be significant (see detailed remarks).

Detailed remarks:

Line 83f: “Reaction was terminated by adding 8 mL of 5% Na2S2O3 [20] for every second 15 min of exposure.” - sentence unclear, please clarify.

Line 89: Please include culture time and conditions for strain Hansenula mrakii

Line 98: “…containing 0.5 g of glass bean…” – do you mean glass beads?

Line 111: “Yeast cells (5 g) …” – please state if this was measured in dry weight.

Line 170: “The injection volume was 10 uL” – Please state what was injected (I suppose untreated wine)

Line 185: “Total tannin content (TTC) in wine was determined according to the method of [24].” – Please write out the name of the author”

Line 203: “… fermented with wild S. cerevisiae…” – The term wild is misleading as the underlying strain purposely is a domesticated fermentation isolate. Consider rephrasing.

Line 205f: “All tested samples were adjusted to 3 Brix…” – Please critically reflect if this could have biased the sensory panel. Since the fermented wines had different alcohol levels, different dilution could have also diluted color and flavor differently. If necessary, include this reflection in the manuscript.

Line 227: Please state how many clones were picked and individually evaluated from the mutated population, and how it was decided to continue with M1, M7, CM3, CM5, CM8.

Line 290: “… both CM8 wines got more panelists for high scores than did SC wines.” – This statement is rather imprecise and depends on the definition of high scores. E.g., more panelists gave the point 8 to SC wines than to CM8 wines

Line 302ff: “Conclusively, CM8-WIN, the topped one rich in total phenolics, tartaric esters, flavonols, anthocyanins, and tannins contents, displayed the greatest colorimetric a*, b*, color density, and highest consumer acceptance in color, flavor, aroma, overall, AI and consumer preference among the wines examined.” - Besides not true for all statements (e.g., anthocyanins), many of these findings were not found significant before.

Line 359: “… in contrast to 173 commercial red wines [5]…” – The PCA in the cited study is only based on 78 wines.

Line 362ff: In my opinion, comparison between the two PCAs is not informative as the data basis for the presented PCA is much smaller, coming from only two grape and two yeast varieties, whereas the PCA in publication 5 is an average of many different conditions from one completely different wine growing region (British Columbia in Canada). The origin of other wine samples should also be mentioned in the publication.

Line 374ff: “All these differences could be reasonably linked to the effects of CM8 high-M mannoproteins by interactions with wine compositions (tannins, flavonols, anthocyanins, tartaric acid, and aroma), especially tannins.” – This statement is false. As demonstrated before (e.g., Figure 3), attributes like phenolics, tartaric acids, etc., contribute equally to CM8-WIN and SC-WIN and are therefore grape dependent.

Line 377f: “the promising effects of enriched high-M mannoproteins from CM8 would be responsible for the high quality attributes…” – Please change would to could.

Line 379f: “which possessed greater TTC, TFC, and slightly higher Aace (total anthocyanins)…” – It was shown that the effect on TFC is not significant.

Line 383f: “Mannoproteins-tannins interactions seem to be remarkable in this study…” – Please precise the term “remarkable”.

Line 388f: “This is further confirmed by the observation that CM8 wine mashes showed precipitates much less than did the SC in the end of fermentation.” – It is not clear where the data for this statement is presented.

Line 390f: “… using a wine starter of higher-M mannoproteins (> 80% mannose), like S. cerevisiae CM8, is more effective than those using wild S. cerevisiae strains…” – Please precise the term “more effective”.

Line 397ff: “… the resultant wines appeared to be of significantly greater total tannins, flavonols, and total anthocyanins levels, higher color, higher flavor, and higher consumer preference for CM8-WIN than for SC-WIN wines.” – flavonol levels were before found to not be significantly different. Also, the term consumer preference is misleading in this case. Overall preference was found to be not significantly different. The acceptance index was higher for CM8 wines, however, it was determined in a way that didn’t allow for significance testing.

Line 404: “Conclusively, using CM8 as a starter is feasible to improve wine quality…” – Please state that this is also dependent on the grape variety as shown in this study.

Reviewer 3 Report

Dear authors,

here are some remarks on your study:

-check the spaces and exponents in the whole text

-row 19 then

-row 23 high-quality

-row 44 extract

Table 2. can you provide a concentration of alcohol in (% vol.)

I have some concerns about the production of wine- it is not clear how many repetitions did you made? (tables say n=3?), because it is written that you used 1.0 kg of grapes -> than you filled 5L jar? ( impossible)-> then 73! consumers tested those wines according to 2 methods (even 5L of wine was not enough)...

Then you say that MLF was conducted at 4°C -> I wonder which bacteria was used at so low temperature?

Round 2

Reviewer 3 Report

Dear authors,

I appreciate the changes you made according to my advice,

best regards